# Leptin Promotes Vasculogenic Mimicry in Breast Cancer Cells by Regulating Aquaporin-1

**DOI:** 10.3390/ijms25105215

**Published:** 2024-05-10

**Authors:** Deok-Soo Han, Eun-Ok Lee

**Affiliations:** 1Department of Science in Korean Medicine, College of Korean Medicine, Graduate School, Kyung Hee University, 26, Kyungheedae-ro, Dongdaemun-gu, Seoul 02447, Republic of Korea; ejr0957@khu.ac.kr; 2Department of Cancer Preventive Material Development, College of Korean Medicine, Graduate School, Kyung Hee University, 26, Kyungheedae-ro, Dongdaemun-gu, Seoul 02447, Republic of Korea

**Keywords:** leptin, vasculogenic mimicry, breast cancer cells, aquaporin-1, Ob-R, STAT3

## Abstract

Leptin is an obesity-related hormone that plays an important role in breast cancer progression. Vasculogenic mimicry (VM) refers to the formation of vascular channels lined by tumor cells. This study aimed to investigate the relationship between leptin and VM in human breast cancer cells. VM was measured by a 3D culture assay. Signal transducers and activators of transcription 3 (STAT3) signaling, aquaporin-1 (AQP1), and the expression of VM-related proteins, including vascular endothelial cadherin (VE-cadherin), twist, matrix metalloproteinase-2 (MMP-2), and laminin subunit 5 gamma-2 (LAMC2), were examined by Western blot. AQP1 mRNA was analyzed by a reverse transcriptase-polymerase chain reaction (RT-PCR). Leptin increased VM and upregulated phospho-STAT3, VE-cadherin, twist, MMP-2, and LAMC2. These effects were inhibited by the leptin receptor-blocking peptide, Ob-R BP, and the STAT3 inhibitor, AG490. A positive correlation between leptin and AQP1 mRNA was observed and was confirmed by RT-PCR. Leptin upregulated AQP1 expression, which was blocked by Ob-R BP and AG490. AQP1 overexpression increased VM and the expression of VM-related proteins. AQP1 silencing inhibited leptin-induced VM and the expression of VM-related proteins. Thus, these results showed that leptin facilitates VM in breast cancer cells via the Ob-R/STAT3 pathway and that AQP1 is a key mediator in leptin-induced VM.

## 1. Introduction

The inhibition of tumor angiogenesis in anticancer therapy has received considerable attention because of its role in promoting growth and metastasis by supplying oxygen and nutrients [1,2]. Strategies that disrupt the function of endothelial cells (ECs) have proven effective in inhibiting tumor angiogenesis [3,4]. However, the anti-cancer efficacy of drugs targeting ECs is often limited [5], as alternative pathways provide blood flow within the tumor [6,7]. A representative phenomenon is vasculogenic mimicry (VM), which was first observed in aggressive skin cancer cells in 1999 [8,9]. VM refers to the formation of vascular channels lined by the tumor cells without ECs [10,11]. It provides nutrients and oxygen to the tumor cells, contributing to tumor growth, invasion, and metastasis [12,13]. VM is a mechanism by which tumors can grow and spread even in the presence of drugs that inhibit angiogenesis within the tumor, and it is emerging as a new therapeutic target for cancer treatment. Among the 20 types of cancer, breast cancer patients with VM have a significantly lower survival rate than those with other types of VM [14]. Therapeutics that specifically target VM to interfere with the formation and function of these tumor cell-derived channels are expected to hinder tumor progression and improve clinical outcomes in patients with breast cancer.

Leptin is an obesity-related hormone secreted from the adipose tissue that manages energy balance, neuroendocrine and immune system operations, and glucose, lipid, and bone metabolism [15]. Obesity is recognized as an important contributing factor to numerous types of cancers, including breast cancer, which is associated with increased leptin secretion [16,17]. Leptin is also secreted by cancer cells, and its overexpression is particularly observed in breast cancer cells [18]. The secreted leptin binds to the leptin receptor (Ob-R) of breast cancer cells and acts as a growth factor, inducing proliferation, survival, angiogenesis, and metastasis of breast cancer cells, thereby promoting the development and progression of breast cancer. This particularly affects cancer recurrence and survival rates in obese postmenopausal women [17,18,19]. 

Aquaporin-1 (AQP1) functions as a transport protein that forms water channels in cell membranes, facilitating the rapid movement of water molecules [20,21]. Some studies have shown that overexpression of AQP1 is associated with the progression of breast cancer [22,23,24]. AQP1 is crucial for regulating water transport by the tumor cells and affects tumor vascular formation and migration [25,26]. Moreover, elevated levels of AQP1 can enhance the invasive and metastatic potential of tumor cells, potentially inducing VM [27,28]. 

Although leptin-induced signaling is related to the synthesis of substances involved in VM, few studies have reported that leptin [29] and Ob-R [30] induce VM. The interaction between leptin and AQP1 can cause changes in water regulation within blood vessels, influencing the formation of vascular structures related to VM [31]. Therefore, this study aimed to investigate the mechanisms by which leptin induces VM in human breast cancer cells. 

## 2. Results

### 2.1. Leptin Promotes VM in Human Breast Cancer Cells

A three-dimensional culture VM formation assay was performed to determine the potential of leptin to form vascular channels by cancer cells. MDA-MB-231 and Hs 578T cells were treated with different concentrations of leptin for 16 h. Leptin led to the dramatic formation of vascular channels in both cell lines in a dose-dependent manner (Figure 1A,B). To explore the mechanism by which VM is induced by leptin, the expression levels of leptin signal- and VM-related proteins were detected by Western blot. After treatment of both cell lines with different concentrations of leptin for 30 min, the phosphorylation of signal transducers and activators of transcription 3 (STAT3) was increased without a change in the STAT3 levels (Figure 1C,D). The protein levels of VE-cadherin, twist, MMP-2, and LAMC2 were upregulated after leptin treatment for 24 h (Figure 1E,F). Thus, these results demonstrate that leptin induces VM in human breast cancer cells.

### 2.2. Leptin Promotes VM through the Ob-R/STAT3 Pathway in Human Breast Cancer Cells

To further confirm the role of Ob-R and STAT3 signaling in leptin-induced VM, MDA-MB-231 and Hs 578T cells were treated with leptin with and without Ob-R BP or AG490 for 24 h, respectively. OB-R BP interacts with Ob-R and prevents leptin from binding to Ob-R [32,33]. AG490 inhibits the JAK/STAT3 signaling [34]. Blockage of leptin-induced signaling by Ob-R BP contributed to a reduction in leptin-induced VM (Figure 2A,B), STAT3 signaling (Figure 2C,D), and the expression of VM-related proteins (Figure 2E,F). Moreover, AG490 blocked leptin-induced STAT3 phosphorylation (Figure 3A,B), VM (Figure 3C,D), and the expression of VM-related proteins (Figure 3E,F). Thus, the role of Ob-R and STAT3 signaling in leptin-induced VM was confirmed. 

### 2.3. Leptin Upregulates AQP1 Expression through the Ob-R/STAT3 Pathway in Human Breast Cancer Cells

To identify genes that are important mediators of leptin-induced VM, TCGA database analysis was conducted. AQP1 showed a high positive Spearman correlation coefficient (0.503) among the genes co-expressed with leptin in invasive breast carcinoma (Figure 4A). AQP1 mRNA was upregulated after leptin treatment for 24 h in both breast cancer cells by RT-PCR (Figure 4B). At the protein level, leptin effectively upregulated the protein expression of AQP1 in a dose-dependent manner (Figure 4C,D). However, the leptin-induced increase in the AQP1 protein expression was reduced after Ob-R BP (Figure 4E,F) or AG490 treatment (Figure 4G,H). Thus, these results suggest that leptin regulates AQP1 expression through the Ob-R/STAT3 pathway and that AQP1 may be an essential mediator of leptin-induced VM in human breast cancer cells.

### 2.4. AQP1 Is a Key Mediator of Leptin-Induced VM in Human Breast Cancer Cells

To demonstrate a novel functional role of AQP1 in leptin-induced VM, gain- and loss-of-function studies were performed using a CRISPR activation plasmid and siRNA, respectively. First, MDA-MB-231 cells and Hs 578T cells were transfected with the AQP1 CRISPR activation plasmid. AQP1 was effectively overexpressed by the AQP1 plasmid compared to the control plasmid (Figure 5A,B), which led to the stimulation of VM (Figure 5C,D) and the upregulation of VM-related proteins in both cell lines (Figure 5E,F). Thus, AQP1 is involved in leptin-induced VM in breast cancer cells. 

MDA-MB-231 cells and Hs 578T cells were then transfected with AQP1 siRNA to silence AQP1 (Figure 6A,B). Effective silencing of AQP1 inhibited leptin-induced VM in both cell lines (Figure 6C,D). The leptin-induced increase in the expression of VM-related proteins was inhibited by AQP1 silencing (Figure 6E,F). Thus, AQP1 mediates leptin-induced VM in breast cancer cells.

## 3. Discussion

Leptin is an obesity-related hormone secreted by the adipose tissue [15] and has been implicated as a link between obesity and breast cancer [16,17]. Leptin is involved in the proliferation, survival, angiogenesis, and metastasis of breast cancer cells and plays an important role in the development of breast cancer in obese postmenopausal women [17,18,19]. VM implies poor prognosis and the presence or absence of VM is an important indicator for the diagnosis and treatment of patients with breast cancer [35]. The relationship between leptin and VM has not been studied extensively [29]. Therefore, this study aimed to identify the mechanism by which leptin induces VM, to develop a new treatment that can increase efficiency and survival rate of patients with breast cancer. 

This study demonstrated that leptin triggers VM in both breast cancer cells (Figure 1A,B). Breast cancer patients with VM have a lower survival rate compared to those without VM [36,37]. Therefore, VM is a new therapeutic strategy for breast cancer patients, and leptin is a VM inducer that may be an attractive target for breast cancer patients with VM. A previous study reported that leptin activates the STAT3 pathway to include VM [29]. STAT3 is a transcription factor that affects breast cancer progression and chemoresistance by regulating several oncogenes [38] and participates in the leptin/Ob-R signaling pathway [39,40]. Leptin plays a vital role in the development and progression of breast cancer by activating STAT3 signaling [41]. In this study, leptin phosphorylated STAT3 (Figure 1C,D). To confirm the involvement of Ob-R and STAT3 signaling in leptin-induced VM, cells were treated with Ob-R BP or AG490, respectively. Leptin-induced VM was markedly blocked in both breast cancer cells treated with these inhibitors (Figure 2A,B and Figure 3C,D). Previous studies have demonstrated that VM was formed through the twist/VE-cadherin/MMP-2/LAMC2 cascade [42,43,44]. Leptin also upregulated the expression of the VM-related proteins (Figure 1E,F). However, these effects were suppressed after treatment with Ob-R BP (Figure 2E,F) and AG490 (Figure 3E,F). These results indicate that leptin has the potential to cause VM through the OB-R/STAT3 pathway. 

TCGA data analysis was conducted to identify a target gene that strongly interacts with leptin. AQP1 was positively correlation with leptin in invasive breast carcinoma (Figure 4A), showing that leptin is associated with AQP1. AQP1 regulates the movement of water and other small molecules across cell membranes [45]. AQP1 induces tumor proliferation and metastasis [46]. Breast cancer tumor cells actively utilize water for angiogenesis and AQP1 plays a vital role in water transport by tumor cells [47]. This study revealed for the first time that leptin regulates the expression of AQP1 at both mRNA (Figure 4B) and protein levels (Figure 4C,D). Leptin-upregulated AQP1 protein expression was inhibited by treatment with OB-R BP (Figure 4E,F) and AG490 (Figure 4G,H), indicating that AQP1 is upregulated by the leptin/Ob-R/STAT3 pathway. High AQP1 expression affects the cellular processes including cell migration, proliferation, and angiogenesis [48]. Overexpression of AQP1 using the activation plasmid (Figure 5A,B) effectively increased VM (Figure 5C,D) and upregulated the expression of VM-related proteins (Figure 5E,F), indicating that AQP1 itself could induce VM. To explore whether APQ1 mediates leptin-induced VM, breast cancer cells were treated with leptin after transfection with AQP1 siRNA (Figure 6A,B). AQP1 silencing inhibited leptin-induced VM (Figure 6C,D) and reduced the expression of VM-related proteins (Figure 6E,F). These results indicate that AQP1 is required for leptin-induced VM. Inhibition of AQP1 contributes to a reduction of VM, thereby suppressing tumor growth [27]. Therefore, leptin and AQP1 are promising targets to inhibit VM in breast cancer cells.

Thus, this study demonstrates the role of leptin in the VM in breast cancer cells. These results are summarized in Figure 7. Leptin upregulates AQP1 expression through the Ob-R/STAT3 pathway, thereby increasing the expression of VM-related proteins and inducing VM. However, more research is needed to determine how leptin regulates AQP1 expression. These results suggest that leptin is an essential target for the prevention and treatment of patients with breast cancer in obese postmenopausal women. Moreover, leptin and AQP1 may be potential biomarkers in VM-related breast cancer. 

## 4. Materials and Methods

### 4.1. Cell Culture

Human triple-negative breast cancer cell lines MDA-MB-231 and Hs 578T were sourced from the Korean Cell Line Bank (KCLB, Seoul, Republic of Korea), and were cultured in RPMI 1640 (Cat: LM 011-01, WELGENE, Daegu, Republic of Korea) and DMEM (Cat: LM 001-05, WELGENE), respectively, supplemented with 10% fetal bovine serum (FBS) (Cat: S101-07, WELGENE) and 1% antibiotics (Cat: LS203-01, WELGENE). Both cell lines were maintained in a humidified incubator at 37 °C and 5% CO_2_.

### 4.2. Three-Dimensional (3D) Culture VM Tube Formation Assay

To evaluate the formation of vascular channels lined by tumor cells, a 3D culture assay was performed on a 24-well plate coated with 100 μL of Matrigel [44]. MDA-MB-231 cells (2.5 × 10^5^) and Hs 578T cells (3.5 × 10^5^) cells were treated with leptin with and without Ob-R BP (Santa Cruz, Danvers, MA, USA) or AG490 (Calbiochem, Darmstadt, Germany) for 16 h. To examine the gain- and loss-of-function of AQP1, a 3D culture assay was performed after transfection using CRISPR activation plasmid or siRNA, respectively. The VM structures formed were visualized using the Ts2_PH in a light microscope at 40× magnification and counted.

### 4.3. Western Blot Analysis 

Cell lysates were subjected to SDS-PAGE (8–12%) separation under a constant voltage range of 60–85 V and then transferred onto membranes (Pall Corporation, Port Washington, NY, USA) at 100 V for 70–90 min. After blocking with 5% skim milk or bovine serum albumin (BSA) for 90 min, the membrane was probed with primary antibodies targeting AQP1, VE-cadherin, twist, phospho-STAT3, STAT3, MMP-2, LAMC2, and β-actin (Table 1) at 4 °C for 24 h. Following incubation, the membrane was probed with a specific secondary antibody for 2 h at room temperature (RT). Each protein band was visualized using an enhanced chemiluminescent reagent (GE Healthcare, Chicago, IL, USA) and quantified using the ImageJ 1.40g software (National Institute of Health, Bethesda, MD, USA).

### 4.4. Isolation of RNA and Reverse Transcriptase-Polymerase Chain Reaction (RT-PCR)

Total RNA extraction was performed using TRIzol reagent (Invitrogen, Carlsbad, CA, USA), and subsequent cDNA synthesis and PCR were conducted as previously described [42,44] using specific primers (Table 2). PCR products were separated by electrophoresis on a 2% agarose gel at 100 V for 40 min.

### 4.5. The Cancer Genome Altas (TCGA) Data Analysis

The co-expression of leptin and AQP1 was analyzed using the TCGA data on the cBioPortal for Cancer Genomics (https://www.cbioportal.org, accessed on 6 March 2023), an open-access resource [49]. The genomic profile was set to mRNA in the invasive breast cancer TCGA dataset (PanCancer Atlas, n = 1084), and then queried by leptin. Among the analyzed results, Spearman’s rank correlation coefficient was used to identify the correlation coefficient.

### 4.6. AQP1 Overexpression by CRISPR Activation Plasmid

MDA-MB-231 cells (1.5 × 10^5^) and Hs 578T cells (2.0 × 10^5^) were seeded on a 6-well plate and transfected with 1 μg control or 0.5 μg AQP1 CRISPR activation plasmid (Santa Cruz) for 48 h using the UltraCruz transfection reagent (Santa Cruz) according to the manufacturer’s protocol. 

### 4.7. AQP1 Silencing by Small Interfering RNA (siRNA)

MDA-MB-231 cells (1.5 × 10^5^) and Hs 578T cells (2.0 × 10^5^) were seeded on a 6-well plate and transfected with 5 nM control or AQP1 siRNA (Santa Cruz) for 48 h using the INTERFERin transfection reagent (Polyplus Transfection, New York, NY, USA) according to the manufacturer’s protocol. 

### 4.8. Statistical Analysis 

The results were reported as the mean ± standard deviation (SD) from three independent experiments. Statistical significance was determined by a Student’s *t*-test (*p* < 0.05) using the GraphPad Prism software (version 5, GraphPad Software Inc., Boston, MA, USA).

## Figures and Tables

**Figure 1 ijms-25-05215-f001:**
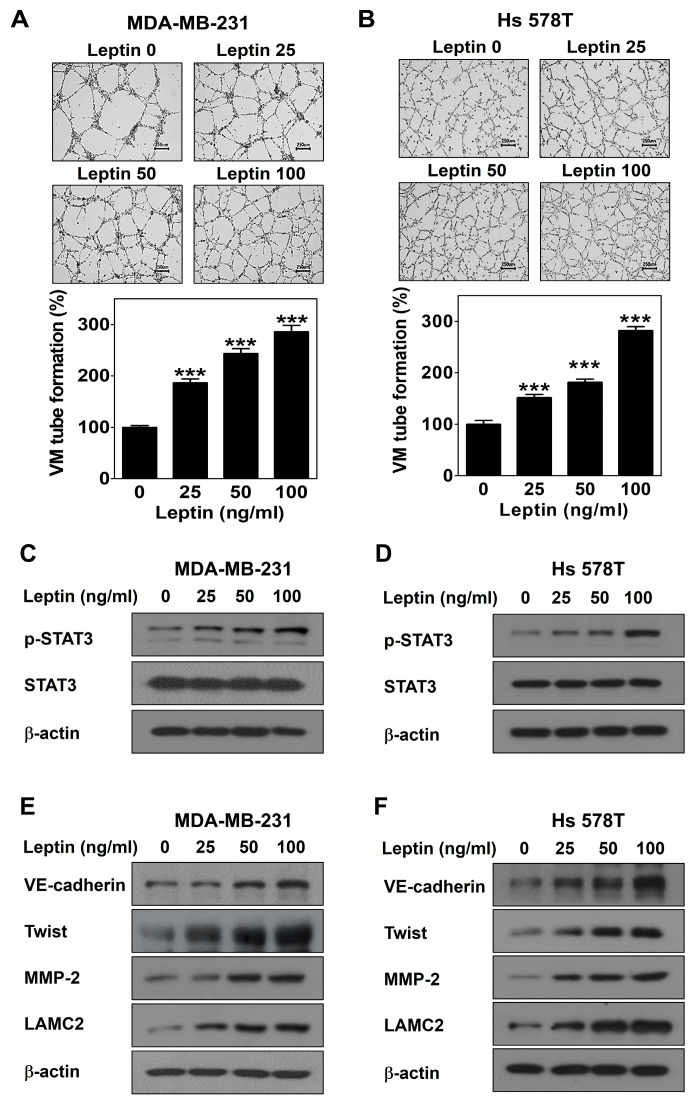
Leptin promotes VM in human breast cancer cells. VM tube formation assay was carried out in MDA-MB-231 cells (**A**) and Hs 578T cells (**B**) treated with leptin. After 16 h of incubation, images were visualized (40× magnification; scale bar = 250 μm) and the number of VM structures formed was counted. Western blot was performed using specific antibody in MDA-MB-231 cells (**C**,**E**) and Hs 578T cells (**D**,**F**) treated with leptin for 30 min (**C**,**D**) or 24 h (**E**,**F**). Data are reported as the mean ± SD and statistical significance is indicated as *** *p* < 0.001 vs. untreated control.

**Figure 2 ijms-25-05215-f002:**
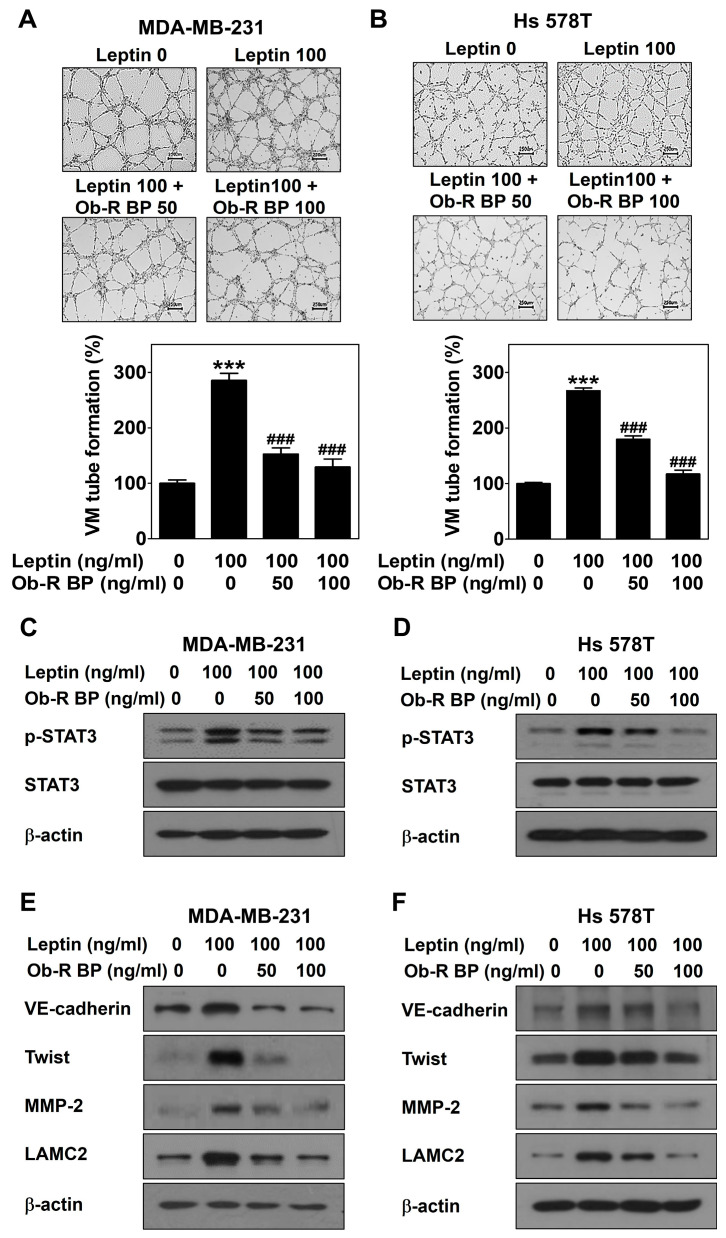
Leptin promotes VM through Ob-R in human breast cancer cells. VM tube formation assay was performed in MDA-MB-231 cells (**A**) and Hs 578T cells (**B**) treated with leptin with and without Ob-R BP. After 16 h of incubation, images were visualized (40× magnification; scale bar = 250 μm) and the number of VM structures formed was counted. Western blot was performed using specific antibodies in MDA-MB-231 cells (**C**,**E**) and Hs 578T cells (**D**,**F**) treated with leptin with and without Ob-R BP for 30 min (**C**,**D**) or 24 h (**E**,**F**). Data are reported as the mean ± SD and statistical significance is indicated as *** *p* < 0.001 vs. untreated control; ### *p* < 0.001 vs. leptin-treated control.

**Figure 3 ijms-25-05215-f003:**
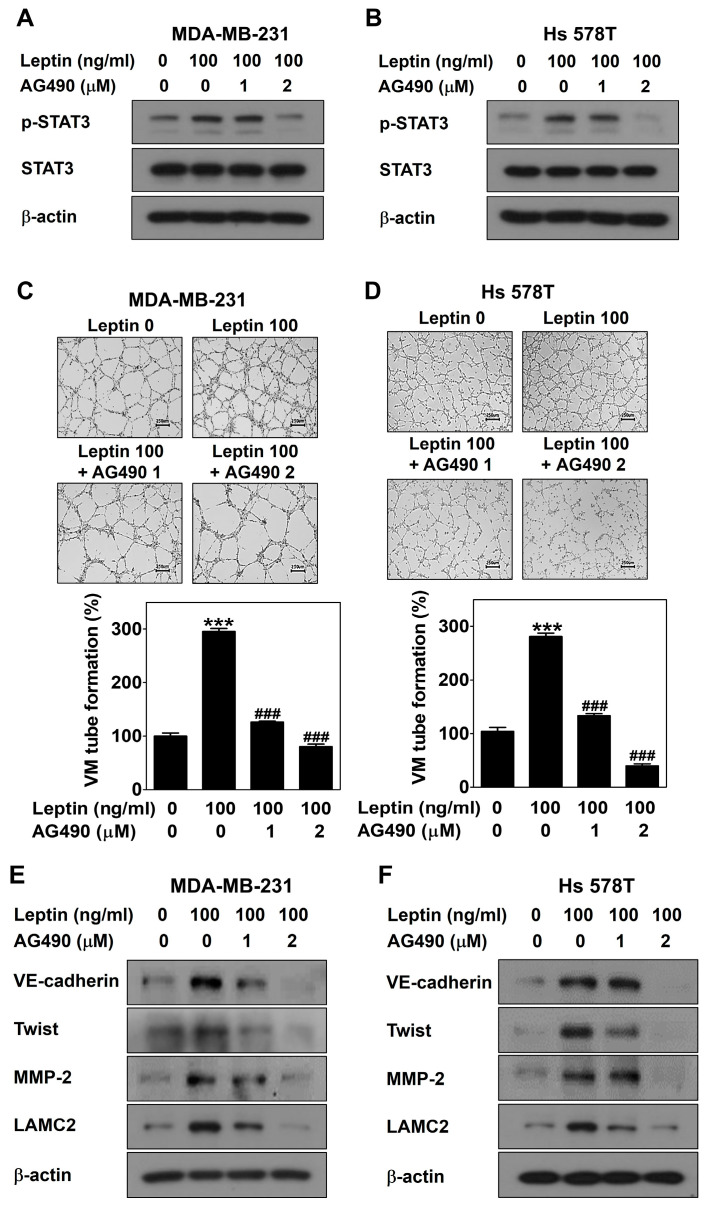
Leptin promotes VM through STAT3 signaling in human breast cancer cells. Western blot was performed using specific antibodies in MDA-MB-231 cells (**A**,**E**) and Hs 578T cells (**B**,**F**) treated with leptin with and without AG490 for 30 min (**A**,**B**) or 24 h (**E**,**F**). VM tube formation assay was carried out in MDA-MB-231 cells (**C**) and Hs 578T cells (**D**) treated with leptin with and without AG490 on a Matrigel-coated well plate. After 16 h of incubation, images were visualized (40× magnification; scale bar = 250 μm) and the number of VM structures formed was counted. Data are reported as the mean ± SD and statistical significance is indicated as *** *p* < 0.001 vs. untreated control; ### *p* < 0.001 vs. leptin-treated control.

**Figure 4 ijms-25-05215-f004:**
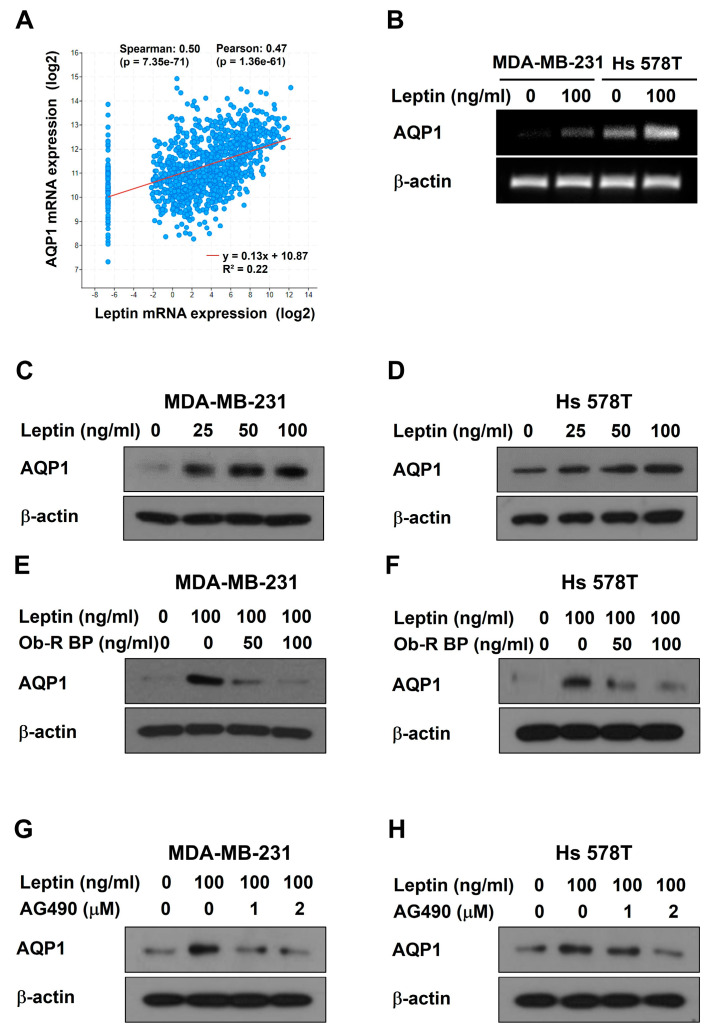
Leptin upregulates AQP1 expression through the Ob-R/STAT3 pathway in human breast cancer cells. (**A**) Co-expression of leptin and AQP1 mRNA was analyzed using the TCGA database. (**B**) AQP1 mRNA expression was confirmed by RT-PCR in breast cancer cells treated with leptin for 24 h. Western blot was performed using specific antibodies in MDA-MB-231 cells (**C**,**E**,**G**) and Hs 578T cells (**D**,**F**,**H**) treated with leptin alone (**C**,**D**), leptin with and without Ob-R BP (**E**,**F**), or leptin with and without AG490 (**G**,**H**) for 24 h.

**Figure 5 ijms-25-05215-f005:**
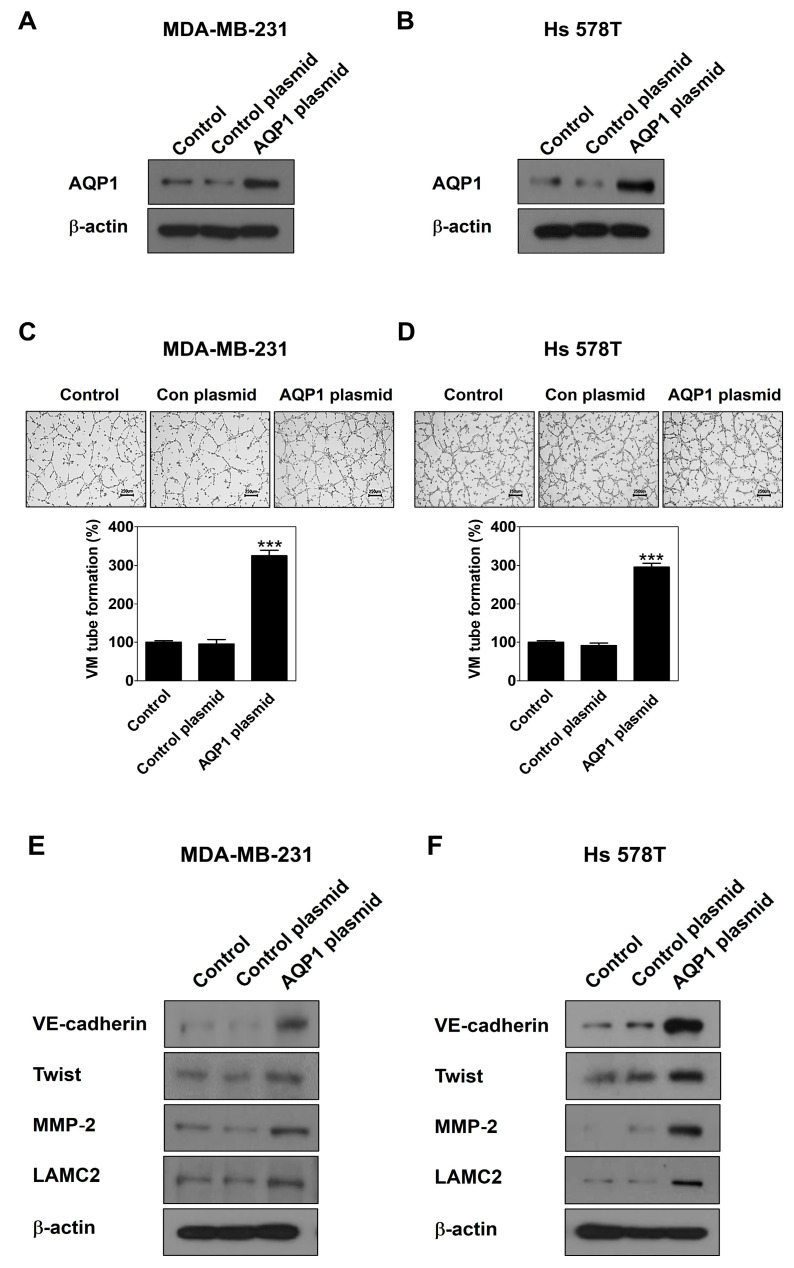
AQP1 overexpression induces VM in human breast cancer cells. Western blot was performed using specific antibodies in AQP1 activation plasmid-transfected MDA-MB-231 cells (**A**) and Hs 578T cells (**B**) for 24 h. VM tube formation assay was carried out in AQP1 activation plasmid-transfected MDA-MB-231 cells (**C**) and Hs 578T cells (**D**). After 16 h of incubation, images were visualized (40× magnification; scale bar = 250 μm) and the number of VM structures formed was counted. Western blot was performed using specific antibodies in AQP1 activation plasmid-transfected MDA-MB-231 cells (**E**) and Hs 578T cells (**F**) for 24 h. Data are reported as the mean ± SD and statistical significance is indicated as *** *p* < 0.001 vs. control plasmid.

**Figure 6 ijms-25-05215-f006:**
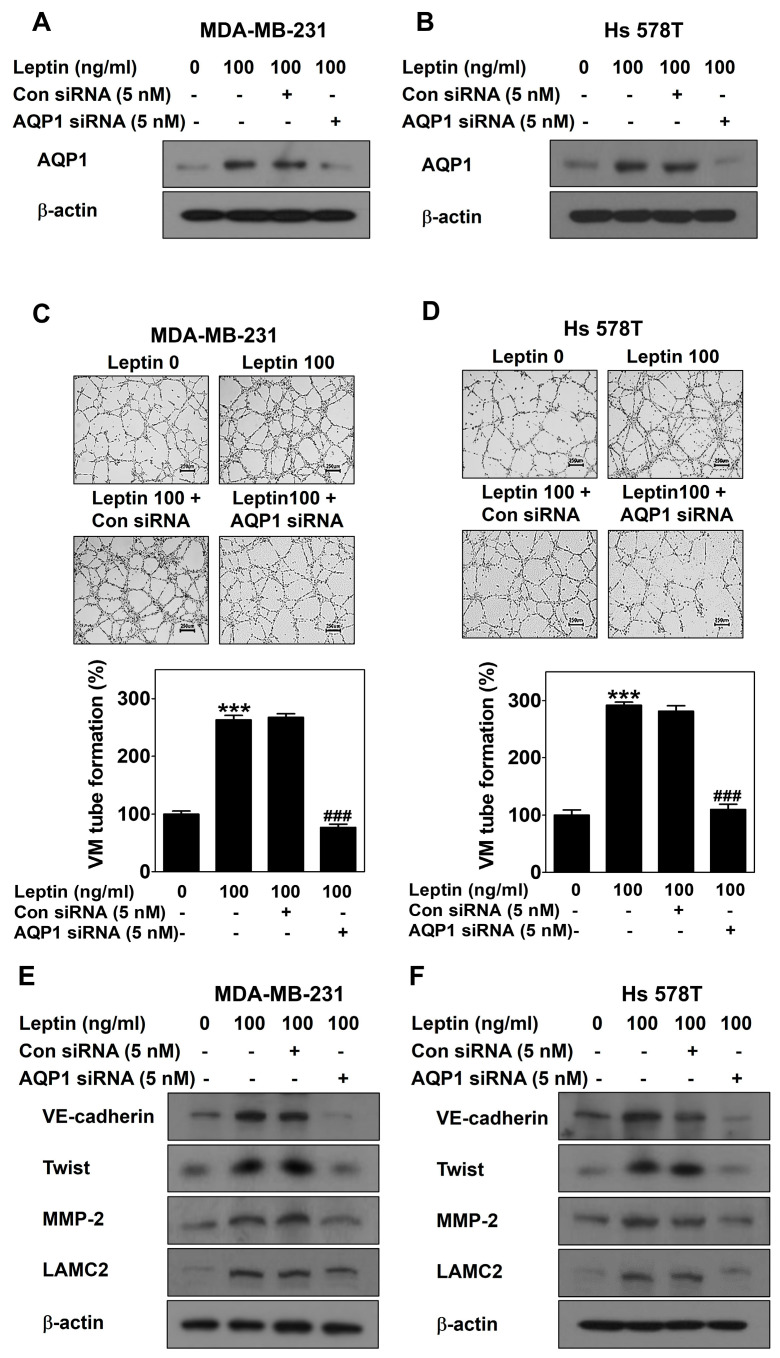
AQP1 silencing inhibits leptin-induced VM in human breast cancer cells. Western blot was performed using specific antibodies in AQP1 siRNA-transfected MDA-MB-231 cells (**A**) and Hs 578T cells (**B**) treated with leptin for 24 h. VM tube formation assay was carried out in AQP1 siRNA-transfected MDA-MB-231 cells (**C**) and Hs 578T cells (**D**) treated with leptin. After 16 h of incubation, images were visualized (40× magnification; scale bar = 250 μm) and the number of VM structures formed was counted. Western blot was performed using specific antibodies in AQP1 siRNA-transfected MDA-MB-231 cells (**E**) and Hs 578T cells (**F**) treated with leptin for 24 h. Data are reported as the mean ± SD and statistical significance is indicated as *** *p* < 0.001 vs. untreated control; ### *p* < 0.001 vs. control siRNA.

**Figure 7 ijms-25-05215-f007:**
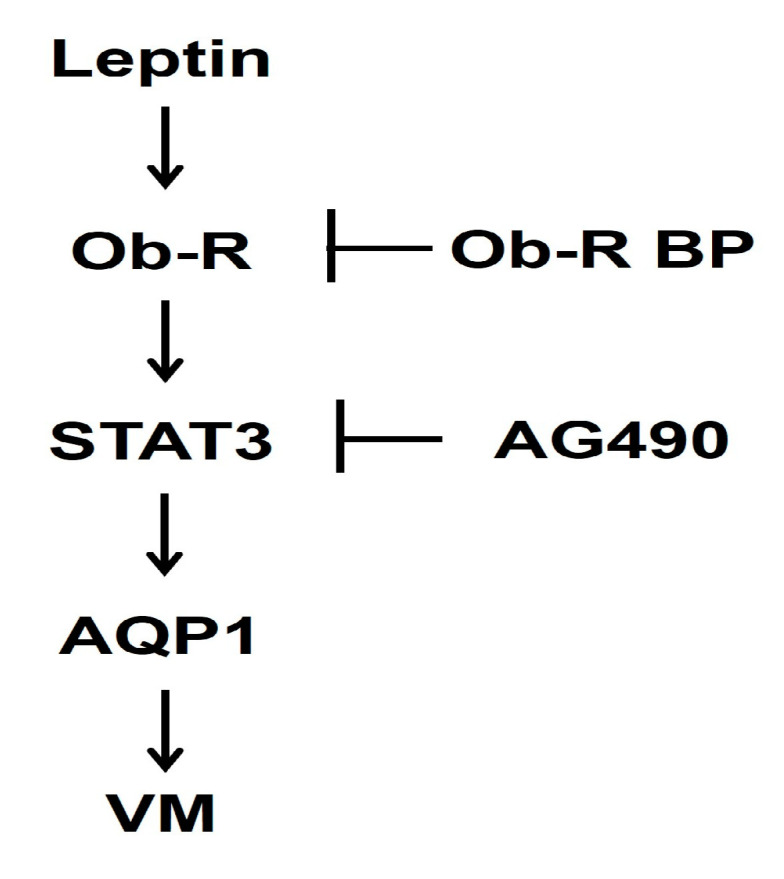
Molecular mechanisms proposed for the leptin-induced VM in human breast cancer cells.

**Table 1 ijms-25-05215-t001:** Antibodies used in this study.

Antibody	Company	Dilution	Product No.
AQP1	Santa Cruz	1:500	SC-25287
β-actin	Sigma-Aldrich	1:20,000	A5316
p-STAT3	CST	1:1000	9145
STAT3	CST	1:5000	12640
MMP-2	Abcam	1:1000	ab86607
LAMC2	Abcam	1:1000	ab96327
VE-cadherin	Abgent	1:1000	AP2724a
Twist	Abcam	1:1000	ab50887
goat anti-rabbit IgG-HRP	CST	1:5000	7074P2
goat anti-mouse IgG-HRP	Bio-Rad	1:5000	STAR120P

Santa Cruz (Danvers, MA, USA); CST, Cell Signaling Technology (Beverly, MA, USA); Sigma-Aldrich (St Louis, MO, USA); Abcam (Cambrige, UK); Abgent (San Diego, CA, USA). Bio-Rad (Hercules, CA, USA).

**Table 2 ijms-25-05215-t002:** Primers used in this study.

mRNA	Primer Sequences	Size	Annealing Temperature
β-actin	S: GAGAAGATGACCCAGATCATGT AS: ACTCCATGCCCAGGAAGGAAGG	463	60
AQP1	S: CAGCCCAAGGACAGTTCAGAG AS: CCATCATGGCTAAGTGCACAG	118	60

## Data Availability

Data is contained within the article.

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
