# Peer review of "Leptin Promotes Vasculogenic Mimicry in Breast Cancer Cells by Regulating Aquaporin-1"

_ijms, 2024, doi:10.3390/ijms25105215_

Round 1

Reviewer 1 Report

Comments and Suggestions for Authors

In this manuscript, authors found leptin induced vasculogenic mimicry (VM) in breast cancer cells through activation of STAT3 signaling and aquaporin-1 (AQP1), accompany with the expression of VM-related proteins, including vascular endothelial cadherin (VE-cadherin), twist, matrix metalloproteinase-2 (MMP-2), and laminin subunit 5 gamma-2 (LAMC2). These effects were inhibited by the leptin receptor- blocking peptide, Ob-R BP, or the STAT3 inhibitor, AG490. A positive correlation between leptin and AQP1 mRNA was observed and was confirmed by RT-PCR. Leptin upregulated AQP1 expression, which was blocked by Ob-R BP and AG490. AQP1 overexpression increased VM and the expression of VM-related proteins. AQP1 silencing inhibited leptin-induced VM and the expression of VM-related proteins. The study demonstrated that leptin facilitates VM in breast cancer cells via the Ob-R/STAT3 pathway and that AQP1 is a key mediator in leptin-induced VM.

1)    Figure 1A&B, 2A&B.. Quantification of VM formation is not appropriate, proper quantification for VM formation needs to be down.

2)    To keep consistency, the unit of leptin or other treatment needs to be included in all figures.

3)    Role of AG490 needs to be included in the results section, instead of in discussion. Figure 3 C-F, what does it mean AG490 1 and 2?

4)    How leptin activates AQP1? Any change of AQP1 mRNA level?

5)    The current discussion was almost the repeat of results. The discussion is about the new findings, strength and weakness of study, any therapeutic potential and future directions.

Comments on the Quality of English Language

The quality of English in this manuscript needs minor improve. 

Author Response

1)    Figure 1A&B, 2A&B.. Quantification of VM formation is not appropriate, proper quantification for VM formation needs to be down.

Response) The degree of VM formation was calculated by counting the number of perfectly connected circles, and considering the control group as 100% and converting it to % of the experimental group.

2)    To keep consistency, the unit of leptin or other treatment needs to be included in all figures.

Response) According to your comments, we have revised it.

3)    Role of AG490 needs to be included in the results section, instead of in discussion.

Response) According to your comment, we have revised this content in the results section.

Figure 3 C-F, what does it mean AG490 1 and 2?

Response) The number is the concentration of AG490 and the unit is mM. This information was included in all figures.

4)    How leptin activates AQP1? Any change of AQP1 mRNA level?

Response) We confirmed that leptin increased the expression of AQP1 mRNA (Figure 4B) and protein (Figure 4C and D) in both cells, as shown in Figure 4B-D.

5)    The current discussion was almost the repeat of results. The discussion is about the new findings, strength and weakness of study, any therapeutic potential and future directions.

Response) According to your comments, we have revised it.

We performed English editing of this MS through a company called “Editage”.

Reviewer 2 Report

Comments and Suggestions for Authors

Reviewing the manuscript titled “Leptin promotes vasculogenic mimicry in breast cancer cells by regulating aquaporin-1”, authors investigate the relationship between leptin and vascular mimicry (VM) in human breast cancer cells. It found that leptin increases VM and up-regulates VM-related proteins, which are inhibited by the leptin receptor-blocking peptide Ob-R BP and the STAT3 inhibitor AG490. AQP1 is a key mediator in leptin-induced VM, suggesting that leptin facilitates VM in breast cancer cells via the Ob-R/STAT3 pathway.

This study is quite important because it fills an important gap in our knowledge regarding the action of leptin on aquaporin-1. In my opinion, the paper could be published if some major issues were addressed:

1. More details about the experimental procedures are needed. It is not a simple description of their titles. It should be easy for readers to find the information they need.

2. It would be quite helpful for the reader to figure the mechanisms of action of leptin that you describe in the discussion.

3. Methodology should come before results. Please correct it according to the journal's instructions.

4. In figure 4, points (a) and (b) are misplaced. Please correct.

5. Give more information about the Spearman correlation. What data do we derive from the corresponding chart? It is not so clear.

Comments on the Quality of English Language

Minor editing of the English language is required.

Author Response

  1. More details about the experimental procedures are needed. It is not a simple description of their titles. It should be easy for readers to find the information they need.

Response) Following the writing method recommended by this journal, we briefly described already well-established methods and provided references. However, we have described “The cBioPortal Database Analysis” in detail.

  1. It would be quite helpful for the reader to figure the mechanisms of action of leptin that you describe in the discussion.

Response) According to your comments, we have attached a figure (Figure 7) of leptin's VM regulation mechanism.

  1. Methodology should come before results. Please correct it according to the journal's instructions.

Response) This paper was written using the journal's template. This journal is written in the following order: introduction, results, discussion, and method.

  1. In figure 4, points (a) and (b) are misplaced. Please correct.

Response) We have revised it.

  1. Give more information about the Spearman correlation. What data do we derive from the corresponding chart? It is not so clear.

Response) Spearman's correlation measures the strength and direction of monotonic association between two variables. AQP1 showed a high positive Spearman correlation coefficient (0.503) among the genes co-expressed with leptin. Positive correlation means that as one variable increases the other increases. We treated both cell lines with leptin to confirm this positive correlation and observed that AQP1 mRNA was increased after leptin treatment (Figure 4B). Also, leptin upregulated AQP1 protein expression (Figure 4C and D). Thus, these results demonstrated that leptin regulates AQP1 expression at both mRNA and protein levels.

We performed English editing of this MS through a company called “Editage”.

Round 2

Reviewer 1 Report

Comments and Suggestions for Authors

The discussion is still mainly repeating the results.

Author Response

Thank you for your valuable comments.

According to your comments, we have revised it.

Reviewer 2 Report

Comments and Suggestions for Authors

The revised version of the manuscript finds me in complete agreement. The authors clearly answered all the issues I raised. I unreservedly recommend the publication of the work.

Author Response

Thank you for your valuable comment.